# Lower Platelet-to-Lymphocyte Ratio Was Associated with Poor Prognosis for Newborn Patients in NICU

**DOI:** 10.3390/medicina58101397

**Published:** 2022-10-06

**Authors:** Yanfei Tang, Yiqun Teng, Lingyan Xu, Guangtao Xu, Deqing Chen, Xin Jin, Wanlu Li, Xiuhui Jin, Wen Zhu, Bo Hu, Ruilin Shen, Yuzhang Zhu

**Affiliations:** 1Department of Pediatrics, Municipal Key-Innovative Discipline, The Second Affiliated Hospital of Jiaxing University, Jiaxing 314001, China; 2Forensic and Pathology Laboratory, Provincial Key Laboratory of Medical Electronics and Digital Health, Institute of Forensic Science, Jiaxing University, Jiaxing 314001, China; 3Department of Immunology and Human Biology, University of Toronto, Toronto, ON M4Y 0B9, Canada; 4Department of Pathology and Municipal Key-Innovative Discipline of Molecular Diagnostics, Jiaxing Hospital of Traditional Chinese Medicine, Zhejiang Chinese Medical University, Jiaxing 314001, China; 5Department of Oncology, The Second Affiliated Hospital of Jiaxing University, Jiaxing 314001, China

**Keywords:** platelet-to-lymphocyte ratio, neonatal intensive care unit, risk factor, newborn, hospital mortality, length of ICU stay, length of hospital stay, 90-day mortality

## Abstract

*Background:* Platelet-to-lymphocyte ratio (PLR) is reported to be related to the outcome of intensive care unit (ICU) patients. However, little is known about their associations with prognosis in newborn patients in neonatal ICU (NICU). The aim of the present study was to investigate the prognostic significance of the PLR for newborn patients in the NICU. *Methods:* Data on newborn patients in the NICU were extracted from the Multiparameter Intelligent Monitoring in Intensive Care III (MIMIC III) database. The initial PLR value of blood examinations within 24 h was analyzed. Spearman's correlation was used to analyze the association of PLR with the length of hospital and ICU stays. The chi-square test was used to analyze the association of PLR with mortality rate. Multivariable logistic regression was used to determine whether the PLR was an independent prognostic factor of mortality. The area under the receiver operating characteristic (ROC) curve was used to assess the predictive ability of models combining PLR with other variables. *Results:* In total, 5240 patients were enrolled. PLR was negatively associated with length of hospital stay and ICU stay (hospital stay: ρ = −0.416, *p* < 0.0001; ICU stay: ρ = −0.442, *p* < 0.0001). PLR was significantly correlated with hospital mortality (*p* < 0.0001). Lower PLR was associated with higher hospital mortality (OR = 0.85, 95% CI = 0.75–0.95, *p* = 0.005) and 90-day mortality (OR = 0.85, 95% CI = 0.76–0.96, *p* = 0.010). The prognostic predictive ability of models combining PLR with other variables for hospital mortality was good (AUC for Model 1 = 0.804, 95% CI = 0.73–0.88, *p* < 0.0001; AUC for Model 2 = 0.964, 95% CI = 0.95–0.98, *p* < 0.0001). *Conclusion:* PLR is a novel independent risk factor for newborn patients in the NICU.

## 1. Introduction

The first month of life is the riskiest time for child survival, accounting for approximately 40% of all childhood mortality [1,2]. Each year, 2.6 million neonates die globally, with 75% of neonatal deaths occurring in the first week of life and 99% of deaths occurring in low- and middle-income countries [3]. Premature delivery and birth-related complications (such as birth asphyxia and neonatal sepsis) are considered to be the main causes of neonatal death [4].

In recent years, the neutrophil-to-lymphocyte ratio (NLR), [5] lymphocyte-to-monocyte ratio (LMR) [6] and platelet-to-lymphocyte ratio (PLR) [7] have been found to be independent predictors of prognosis in various benign and malignant conditions [8,9,10,11]. Moreover, NLR, LMR and PLR were reported to be related to the outcome of intensive care unit (ICU) patients, because of their rapid response to systemic inflammation and stress [12,13,14]. NLR, MLR and PLR, in the non-survival group were statistically higher than those in the survival group of sepsis patients, and the cut-off values of NLR, PLR and MLR are 8.66, 275.51 and 0.74%, respectively [15]. However, little is known about the associations of NLR, LMR and PLR with prognosis in newborn patients in the neonatal intensive care unit (NICU). A previous investigation shows that PLR and NLR are identified as the factors to independently predict the risk of early-onset neonatal sepsis (EONS), and the predicting ability of NLR is better than PLR (At the AUC value of NLR ≥ 3.169, the sensitivity of predicting EONS was 77%, and the specificity was 78%) [16]. These data suggest that the expression level of inflammatory indicators varies with the type and severity of the disease.

The primary purpose of this study was to determine the associations of NLR, LMR and PLR with hospital mortality in newborn patients in the NICU. In the present study, we were the first to report that only PLR can serve as an independent risk factor for newborn patients in the NICU.

## 2. Materials and Methods

### 2.1. Data Source

A retrospective cohort study design was used in this study. Data were obtained from the ICU database, a free accessible critical care database of Medical Information Mart for Intensive Care III (MIMIC-III). The clinical data of patients who stayed in the ICU of Beth Israel Deaconess Medical Center (BIDMC) between 2001 and 2012 were selected [17]. The institutional review boards of both the BIDMC and the Massachusetts Institute of Technology Affiliates approved access to the database. No informed consent was required because all of the data were deidentified.

### 2.2. Patient Selection

Clinical data of eligible patients in the MIMIC-III database were selected for analysis in this study. The eligibility criteria were (1) newborn patients admitted to the NICU regardless of diseases; and (2) newborn patients with routine blood examinations within 24 h of admission. Exclusion criteria were newborn patients with trauma or surgery, autoimmune disease, acute or chronic infection, malignant tumor, and pulmonary, hepatic or renal dysfunction admitted to the NICU. In the present study, the initial value of blood examinations within 24 h was analyzed.

### 2.3. Data Extraction

All of the data were obtained and extracted by using the Structured Query Language (SQL), and pgAdmin4 for PostgreSQL was used as the administrative platform. The extracted data mainly included age, sex, birthweight, heart rate (HR), laboratory parameters (red blood cell count (RBC), peripheral white blood cell count (WBC), platelet count, lymphocyte count, neutrophil count, monocytes count), comorbidities (congestive heart failure, cardiac arrhythmias, valvular disease, pulmonary circulation disorder, hypertension, liver disease and renal failure), the Simplified Acute Physiology Score (SAPS) II, the Sequential Organ Failure Assessment (SOFA) score and the model for end-stage liver disease (MELD) score. PLR was calculated by dividing the platelet count by the lymphocyte count. NLR was calculated by dividing the neutrophil count by the lymphocyte count. LMR was calculated by dividing the lymphocyte count by the monocyte count. Since there was little missing data (<1.5%), it was omitted in the further investigation.

### 2.4. Outcome Variables

The following outcome variables were extracted: hospital mortality, length of ICU stay, length of hospital stay and 90-day mortality (post-ICU admission). Because a patient may have had more than one ICU admission during a single hospitalization, the length of ICU stay was entirely determined by the first ICU hospitalization. 

### 2.5. Statistical Analysis

Continuous variables were presented as the mean ± standard deviation or the median (interquartile range) and were compared via *t*-test or the Mann–Whitney U test. Categorical data were presented as numbers with proportions and analyzed via the χ^2^ test. The correlation between length of ICU stay and hospital stay with the laboratory parameters was assessed with the nonparametric Spearman’s rank correlation test. Logistic regression with the univariate and multivariate analyses was used to identify independent prognostic factors of mortality (hospital mortality and 90-day mortality) for newborn NICU patients. Two different models were designed to adjust for potential confounders. Model 1 was adjusted for NLR, LMR, MELD, SAPS II and liver disease. Moreover, Model 2 was adjusted for NLR, LMR and MELD. Receiver operating characteristic (ROC) curves were constructed, and the area under the curve (AUC), sensitivity and specificity were calculated. *p*-values of less than 0.05 were considered to indicate statistical significance.

## 3. Results

### 3.1. Baseline Characteristics of the Study Population

In total, 5240 patients who met the selection criteria were enrolled in our study, among whom 43 patients (0.82%) died in the hospital. The baseline characteristics of the enrolled patients were summarized in Table 1.

The demographic characteristics of the survivors and nonsurvivors were presented in Table 1. No significant differences were observed for age and sex between nonsurvivors and survivors. Nonsurvivors had a lower birthweight and HR. Moreover, nonsurvivors had much lower RBCs, WBCs, neutrophils, platelets, NLRs and PLRs. Nonsurvivors tended to have higher lymphocytes, LMRs, SAPS II scores, SOFA scores and MELD scores, as well as a history of hyaline membrane disease, sepsis, liver disease and renal failure (Table 1).

### 3.2. Association of Inflammatory Markers with Length of Hospital Stay and ICU Stay in Newborn Patients in the NICU

LMR, NLR and PLR were reported to be related to the outcome of various diseases. To investigate the associations of these inflammatory markers with the length of hospital stay and ICU stay in newborn patients in the NICU, Spearman’s rank correlation test was used, and the results were shown in Table 2. LMR was significantly positively associated with length of hospital stay and ICU stay (hospital stay: Spearman’s rho = 0.228, *p* < 0.0001; ICU stay: Spearman’s rho = 0.254, *p* < 0.0001). Both NLR and PLR were negatively associated with length of hospital stay and ICU stay (for NLR, hospital stay: Spearman’s rho = −0.427, *p* < 0.0001; ICU stay: Spearman’s rho = −0.448, *p* < 0.0001. For PLR, hospital stay: Spearman’s rho = −0.416, *p* < 0.0001; ICU stay: Spearman’s rho = −0.442, *p* < 0.0001).

### 3.3. Association of Inflammatory Markers with Hospital Mortality in Newborn Patients in the NICU

In the present study, the correlation of inflammatory markers with mortality in newborn patients in the NICU was investigated. Quartiles of LMR, NLR and PLR were significantly correlated with hospital mortality (all *p* < 0.0001) (Table 3). A higher rate of hospital mortality was observed in patients in the fourth LMR quartile than in those in the first, second and third quartiles. For NLR and PLR, a higher rate of hospital mortality was observed in patients in the first quartile than in those in other quartiles.

### 3.4. Prognostic Significance of PLR in Newborn Patients in the NICU

As shown in Table 4, PLR, NLR and LMR were associated with hospital mortality and 90-day mortality (PLR for hospital mortality: OR = 0.76, 95% CI = 0.69–0.83, *p* < 0.0001. NLR for hospital mortality: OR = 0.50, 95% CI = 0.37–0.69, *p* < 0.0001. LMR for hospital mortality: OR = 1.03, 95% CI = 1.01–1.05, *p* < 0.0001. PLR for 90-day mortality: OR = 0.76, 95% CI = 0.69–0.84, *p* < 0.0001. NLR for 90-day mortality: OR = 0.49, 95% CI = 0.36–0.68, *p* < 0.0001. LMR for 90-day mortality: OR = 1.04, 95% CI = 1.02–1.05, *p* < 0.0001).

As shown in Table 5, only PLR was significantly associated with hospital mortality (Model 1: OR = 0.85, 95% CI = 0.75–0.95, *p* = 0.005; Model 2: OR = 0.75, 95% CI = 0.67–0.84, *p* < 0.0001) and 90-day mortality (Model 1: OR = 0.85, 95% CI = 0.76–0.96, *p* = 0.010; Model 2: OR = 0.79, 95% CI = 0.71–0.89, *p* < 0.0001) in all models.

### 3.5. Prognostic Predictive Ability of Models Combining PLR with Other Variables for Hospital Mortality in Newborn Patients in the NICU

To evaluate the predictive ability of models combining PLR and other clinical variables for hospital mortality, an ROC curve analysis was performed, and the AUC for Model 1 (Model 1: PLR, LMR and NLR) was 0.804 (95% CI = 0.73–0.88, *p* < 0.0001) and for Model 2 (Model 2: PLR, LMR, NLR, SOFA and MELD) was 0.964 (95% CI = 0.95–0.98, *p* < 0.0001) (Figure 1).

## 4. Discussion

In the present study, we found that LMR was significantly positively associated with length of hospital stay and ICU stay while both NLR and PLR were negatively associated with length of hospital stay and ICU stay. PLR, NLR and LMR were associated with hospital mortality and 90-day mortality, but only PLR was significantly associated with hospital mortality and 90-day mortality in the multivariate analyses. The prognostic predictive ability of models combining PLR with other variables for hospital mortality was good. To our knowledge, this is the first investigation to demonstrate that PLR can serve as an independent risk factor for newborn patients in the NICU.

Inflammatory cells, including WBCs and their subtypes (such as lymphocytes, monocytes and neutrophils), have been well validated to play an indispensable role in various benign and malignant conditions. Moreover, platelets could play a critical role in the immunomodulatory and inflammatory process, Ref. [18] by inducing the release of inflammatory cytokines and interacting with different kinds of immune cells, such as neutrophils, T-lymphocytes, and macrophages (the precursors of macrophages are monocytes), which contribute to the initiation or exacerbation of the inflammatory process [19,20]. Low lymphocyte counts could represent a suppressed immune and inflammatory response, which is related to inflammatory disease [21,22,23]. Thus, PLR was proposed to serve as a novel systematic inflammatory indicator [24,25].

The association between PLR and outcomes was different in different cohorts. Both high and low PLR were associated with increased mortality, among critically ill patients with acute kidney injury (AKI) [26]. In another study, high PLR was positively associated with increased epicardial adipose tissue deposition in diabetes patients [27,28]. Wang et al. showed that high PLR was independently associated with shorter disease-free days and lower overall survival rates in lung adenosquamous carcinoma [29]. For fetal malnutrition, cord-blood PLR negatively correlates with term fetal malnutrition in gestational age neonates [30]. Maternal PLR is negatively correlated with the week of birth and birth weight of the infant [31]. Our data showed that PLR was associated with the prognosis of newborn patients in the NICU. PLR was negatively associated with the length of hospital stay and ICU stay. A higher rate of hospital mortality was observed in patients in the first PLR quartile than in those in the other quartiles. The prognostic predictive ability of models combining PLR with other variables for hospital mortality was good (AUC for Model 1 = 0.804; AUC for Model 2 = 0.964). These data suggest that PLR can serve as an independent risk factor for newborn patients in the NICU. 

## 5. Conclusions

In summary, we demonstrate that lower PLR is significantly associated with higher hospital mortality. The prognostic predictive ability of models combining PLR with other variables for hospital mortality is good. PLR is a novel independent risk factor for newborn patients in the NICU.

## 6. Limitations of the Study

This is a single-center retrospective study. PLR can be time-varying, which was not fully captured in the current analysis. For logistic regression with the univariate and multivariate analyses, models without adjustment for covariates may be biased due to uncertainty or lack of confounders. For example, the interaction or non-linearity for the relationship between covariates and outcome were not considered in this manuscript. To solve this problem, ensemble modeling can address non-linearity automatically without pre-specification in future research. Moreover, the disease definition of the MIMIC III database is based on the ICD-9-CM code. Therefore, some important information is lacking in the database, including cardiac color Doppler ultrasound results.

## Figures and Tables

**Figure 1 medicina-58-01397-f001:**
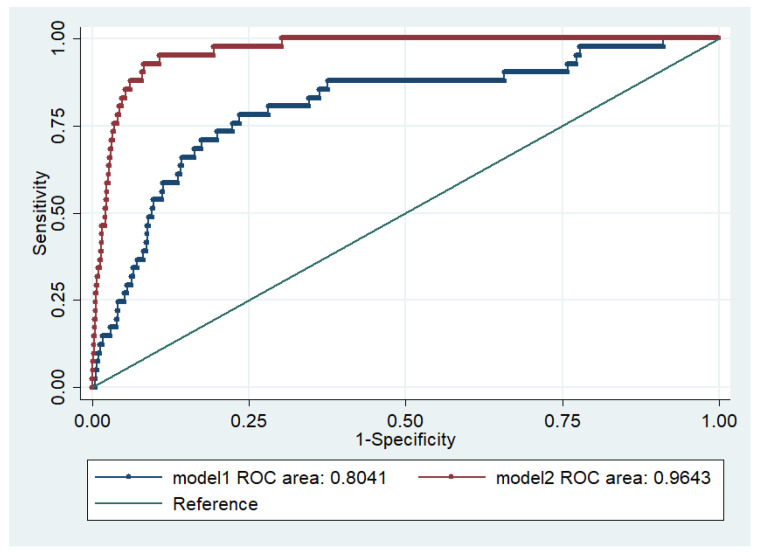
ROC curve for the predictive models for hospital mortality. Note: Model 1 included PLR, LMR and NLR. Model 2 included PLR, LMR, NLR, SOFA and MELD scores. AUC, area under the curve. ROC, receiving operating characteristic.

**Table 1 medicina-58-01397-t001:** Baseline characteristics of the study population with different survival status in hospital.

	Survivors (*n* = 5197)	Nonsurvivors (*n* = 43)	*p* Value
**Demographics**			
Age (d)	0.500 ± 0.004	0.548 ± 0.040	0.311
Male, *n* (%)	2823 (53.87%)	26 (0.50%)	0.420
Female, *n* (%)	2374 (45.31%)	17 (0.32%)	
Birthweight (kg)	1.63 (1.10–2.48)	0.84 (0.62–1.59)	<0.0001
Heart rate (bpm)	152 (140–169)	150 (140–168)	0.001
**Laboratory events**			
RBC, 10^6^/L	4.14 (3.70–4.62)	3.80 (3.45–4.23)	0.001
WBC, 10^9^/L	15.3 (10.9–19.8)	8.8 (4.7–11.4)	<0.0001
Lymphocytes, %	29 (21–42)	54.55 (33–67)	<0.0001
Neutrophils, %	57 (42.9–66)	27 (19–46)	<0.0001
Monocytes, %	7 (4–9.2)	7 (4–11)	0.707
Platelets, 10^9^/L	284 (232–338)	212 (174–240)	<0.0001
LMR	4.33 (2.75–7.4)	7.67 (4.17–13)	<0.0001
NLR	2 (1.02–3.13)	0.55 (0.28–1.64)	0.0008
PLR	9.77 (6.15–14.36)	3.67 (2.73–5.72)	<0.0001
**Comorbidities**			
Hyaline membrane disease	180 (3.56%)	10 (23.81%)	<0.0001
Sepsis	180 (3.56%)	10 (23.81%)	<0.0001
Congestive heart failure	14 (0.27%)	0 (0.00%)	0.736
Cardiac arrhythmias	19 (0.36%)	0 (0.00%)	0.694
Valvular disease	8 (0.15%)	0 (0.00%)	0.799
Pulmonary circulation disorder	5 (0.10%)	0 (0.00%)	0.840
Hypertension	10 (0.19%)	0 (0.00%)	0.773
Liver disease	2 (0.03%)	1 (0.015%)	<0.0001
Renal failure	0 (0.00%)	1 (0.015%)	<0.0001
**Scores**			
SAPS II	18.86 ± 0.15	37.58 ± 0.81	<0.0001
SOFA	2.75 ± 0.05	10.91 ± 0.41	<0.0001
MELD	7.97 ± 0.05	11.64 ± 0.70	<0.0001

Note: Values are presented as the mean ± standard deviation, median (interquartile range), or number of patients (%). LMR, lymphocyte-to-monocyte ratio; MELD, model for end-stage liver disease score; NLR, neutrophil-to-lymphocyte ratio; PLR, platelet-to-lymphocyte ratio; RBC, red blood cell; SOFA, Sequential Organ Failure Assessment score; SAPS II, Simplified Acute Physiology Score II; WBC, white blood cell.

**Table 2 medicina-58-01397-t002:** The correlation of LMR, NLR and PLR with hospital stay and ICU stay.

	Length of Hospital Stay		Length of ICU Stay	
	Spearman’s Rho	*p* Value	Spearman’s Rho	*p* Value
LMR	0.228	<0.0001	0.254	<0.0001
NLR	−0.427	<0.0001	−0.448	<0.0001
PLR	−0.416	<0.0001	−0.442	<0.0001

Note: ICU, intensive care unit; LMR, lymphocyte-to-monocyte ratio; NLR, neutrophil-to-lymphocyte ratio; PLR, platelet-to-lymphocyte ratio.

**Table 3 medicina-58-01397-t003:** The relationship between LMR, NLR and PLR with hospital mortality.

	Q1	Q2	Q3	Q4	*p* Value
LMR					
Survivors	1364 (26.03%)	1288 (24.58%)	1288 (24.58%)	1257 (23.99%)	<0.0001
Nonsurvivors	3 (0.06%)	5 (0.10%)	6 (0.11%)	29 (0.55%)	
NLR					
Survivors	1285 (24.52%)	1277 (24.37%)	1284 (24.50%)	1351 (25.78%)	<0.0001
Nonsurvivors	31 (0.59%)	4 (0.076%)	4 (0.076%)	4 (0.076%)	
PLR					
Survivors	1258 (24.01%)	1283 (24.48%)	1288 (24.58%)	1368 (26.11%)	<0.0001
Nonsurvivors	32 (0.61%)	5 (0.10%)	2 (0.038%)	4 (0.076%)	

Note: LMR, lymphocyte-to-monocyte ratio; NLR, neutrophil-to-lymphocyte ratio; PLR, platelet-to-lymphocyte ratio; Q, Quartiles of LMR, NLR and PLR. %, percentage of survivors or nonsurvivors in each quartile in total patients of total patients. For each indicator, the data in the upper column was for the survivors, and the below for nonsurvivors.

**Table 4 medicina-58-01397-t004:** Univariate logistic regression analyses for prognosis in newborn patients.

Outcome	OR	95% CI	*p* Value
Hospital mortality			
PLR	0.76	0.69–0.83	<0.0001
NLR	0.50	0.37–0.69	<0.0001
LMR	1.03	1.01–1.05	<0.0001
MELD	1.19	1.12–1.25	<0.0001
SAPS II	1.17	1.13–1.21	<0.0001
Liver disease	63.18	5.62–710.58	0.001
90-day mortality			
PLR	0.76	0.69–0.84	<0.0001
NLR	0.49	0.36–0.68	<0.0001
LMR	1.04	1.02–1.05	<0.0001
MELD	1.18	1.12–1.25	<0.0001
SAPS II	1.17	1.13–1.22	<0.0001
Liver disease	63.18	5.62–710.58	0.001

Note: CI, confidence interval; LMR, lymphocyte-to-monocyte ratio; MELD, model for end-stage liver disease score; NLR, neutrophil-to-lymphocyte ratio; OR, odds ratio; PLR, platelet-to-lymphocyte ratio; SAPS II, Simplified Acute Physiology Score II.

**Table 5 medicina-58-01397-t005:** Association between PLR with prognosis of newborn patients.

	Model 1		Model 2	
Variable	OR (95% CI)	*p* Value	OR (95% CI)	*p* Value
Hospital Mortality				
PLR	0.84 (0.75–0.95)	0.005	0.75 (0.67–0.84)	<0.0001
NLR	1.37 (0.98–1.90)	0.063	1.18 (0.85–1.65)	0.315
LMR	1.01 (0.99–1.03)	0.253	1.01 (0.99–1.03)	0.377
MELD	0.89 (0.80–0.99)	0.028	1.11 (1.05–1.19)	<0.0001
SAPS II	1.19 (1.14–1.26)	<0.0001		
Liver disease	11.39 (0.84–154.04)	0.067		
90-day Mortality				
PLR	0.85 (0.76–0.96)	0.010	0.79 (0.71–0.89)	<0.0001
NLR	1.32 (0.94–1.86)	0.106	1.15 (0.82–1.63)	0.417
LMR	1.01 (0.99–1.03)	0.182	1.01 (0.99–1.03)	0.232
MELD	0.88 (0.79–0.98)	0.017	1.12 (1.04–1.19)	0.001
SAPS II	1.20 (1.14–1.27)	<0.0001		
Liver disease	11.53 (0.83–159.82)	0.068		

Note: Model 1 was adjusted for NLR, LMR, MELD, SAPS II and liver disease. Model 2 was adjusted for NLR, LMR and MELD. CI, confidence interval; LMR, lymphocyte-to-monocyte ratio; MELD, model for end-stage liver disease score; NLR, neutrophil-to-lymphocyte ratio; OR, odds ratio; PLR, platelet-to-lymphocyte ratio; SAPS II, Simplified Acute Physiology Score II.

## Data Availability

The datasets presented in this study can be found in online repositories. The names of the repository/repositories and accession number(s) can be found in the article.

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
