# Peer review of "Lower Platelet-to-Lymphocyte Ratio Was Associated with Poor Prognosis for Newborn Patients in NICU"

_medicina, 2022, doi:10.3390/medicina58101397_

Round 1

Reviewer 1 Report

Publication concerns the important problem of death of newborns. PLR could be a simple marker of death risk.

Suggestions of correction:

1. References: number 3 doesn't concern the epidemiology of newborns' death.

2. line 37 "The first month of life is riskest.." should be "The first month of life is the riskest....."

3. What is the source of normal value of cells (monocytes, neutrophiles etc.)? The numbers showed in manuscript are not normal value.

4. Results. Line 140 and 154 - general comments about methods should be in Material and methods.

5. Line 201 - double "was" -one should be deleted.

Author Response

Response to Reviewer 1

Comments and Suggestions for Authors

Publication concerns the important problem of death of newborns. PLR could be a simple marker of death risk.

Suggestions of correction:

  1. References: number 3 doesn't concern the epidemiology of newborns' death.

Response: Thanks for your comments. The number 3 of the references has been deleted as your suggestion.

  1. line 37 "The first month of life is riskest.." should be "The first month of life is the riskest....."

Response: Thanks for your comments. The sentences have been revised as the suggestion, and highlighted in manuscript.

  1. What is the source of normal value of cells (monocytes, neutrophiles etc.)? The numbers showed in manuscript are not normal value.

Response: Thanks for your comments. We found in the literature that the range of CBC was different in different countries, and even in the same country, different testing conditions or different races can lead to different normal value ranges. Therefore, for a more rigorous description, we remove the normal value range in the text.

  1. Results. Line 140 and 154 - general comments about methods should be in Material and methods.

Response: Thanks for your comments. The general comments about the patients has been moved into Material and methods as the suggestion, and highlighted in manuscript.

  1. Line 201 - double "was" -one should be deleted.

Response: Thanks for your comments. The sentences have been revised as the suggestion and highlighted in manuscript.

Reviewer 2 Report

Platelet-to-lymphocyte ratio has been demonstrated in research papers to be related to outcomes of ICU patients. However, this ratio has not been assessed in the NICU population. The purpose of this study is to investigate the prognostic significance of PLR for newborn patients in the NICU. In particular, the authors focused on its relation to mortality rate and length of hospital stay – both of which are important in clinical care of NICU patients. To the authors knowledge, this is a good paper of its kind to demonstrate clinical relevance of this ratio in the NICU and can be easily applied to many patients. The sample size was large which is good for generalizability of the results.

Suggestions:

1.       Title – could use specificity as to what it is an independent risk factor of? For example, lower platelet to lymphocyte ratio prolongs NICU hospitalizations or something like that

2.       Line 37 – “riskiest time for child survival”, needs to be fixed grammatically to convey meaning

3.       Line 55-57 – statement is confusing. Perhaps get rid of first part of sentence and state that expression of inflammatory indicators varies with type and severity of disease.

4.       Line 71-81 – patient selection and eligibility criteria. The exclusion criteria are very vague. What is “pulmonary dysfunction”. Almost all infants admitted to the NICU who are premature have pulmonary dysfunction of some kind. Diagnosis need to be specified as this makes it unclear what type of patients the study did include. These criteria weaken the study

5.       Line 93-95 – This statement can be interpreted to mean that since there was little missing data, then it was omitted. But I believe the authors are trying to say there was little missing data so it was included in analysis or something like this? Needs to be clarified

6.       Table 1 – the demographics included need to include female gender if includes male. In addition, age is not as helpful as gestational age at birth. Heart rate is also not as clinically helpful given it is not clear where this heart rate comes from (just at birth? On average over duration of hospital stay?) and can fluctuate depending on the patients status. If a hemodynamic marker is to be included, mean arterial pressure (MAP) should be included. The Comorbidities contradict the exclusion criteria as hyaline membrane disease is a pulmonary dysfunction disease. Need clarification on what a “pulmonary circulation disorder” is.

7.       Table 3 – this is probably the crux of the paper in graphic form, however it is confusing. The percentages are not clarified in regards to what they mean (% of population? Mortality?) and the trends are not clear. Either need to take away percentages or clarify in the table description.

8.       Line 201 – grammatical error, need to take out second “was”

9.       Line 206 – “hospital mortality was moderately good”. How was moderately good predictive ability determined?

Author Response

Response to Reviewer 2

Comments and Suggestions for Authors

Platelet-to-lymphocyte ratio has been demonstrated in research papers to be related to outcomes of ICU patients. However, this ratio has not been assessed in the NICU population. The purpose of this study is to investigate the prognostic significance of PLR for newborn patients in the NICU. In particular, the authors focused on its relation to mortality rate and length of hospital stay – both of which are important in clinical care of NICU patients. To the authors knowledge, this is a good paper of its kind to demonstrate clinical relevance of this ratio in the NICU and can be easily applied to many patients. The sample size was large which is good for generalizability of the results.

Suggestions:

  1. Title – could use specificity as to what it is an independent risk factor of? For example, lower platelet to lymphocyte ratio prolongs NICU hospitalizations or something like that

Response: Thanks for your comments. We changed the title to “Lower platelet-to-lymphocyte ratio was associated with poor prognosis for newborn patients in NICU” according to the suggestion. We hope the changes meet your requirements. Thank you.

2.Line 37 – “riskiest time for child survival”, needs to be fixed grammatically to convey meaning

Response: Thanks for your comments. We modified the sentence to “The first month of life is the riskest....” according to your and reviewer-1’ comment. The changes has been highlighted in manuscript.

3.Line 55-57 – statement is confusing. Perhaps get rid of first part of sentence and state that expression of inflammatory indicators varies with type and severity of disease.

Response: Thanks for your comments. The sentences have been revised as the suggestion and highlighted in manuscript.

  1. Line 71-81 – patient selection and eligibility criteria. The exclusion criteria are very vague. What is “pulmonary dysfunction”. Almost all infants admitted to the NICU who are premature have pulmonary dysfunction of some kind. Diagnosis need to be specified as this makes it unclear what type of patients the study did include. These criteria weaken the study

Response: Thanks for your comments. The definition of disease in the database was mainly based on the ICD-9-CM code. The diagnosis based on the ICD-9-CM code was strictly determined by trained doctors [1]. These trained doctors reading formal clinical records and nursing records at the end of hospitalization, and data entry based on these records [1]. Some variables such as echocardiography, gestational age, MAP, pre-hospital medication or detailed medical history may be unavailable or contain many missing values. Disease definitions based on specific variables may lead to potential biases. However, for large medical databases or cohorts, the management of variables and events, including diagnoses, procedures, drugs, was mainly based on structural codes. Currently, extracting diagnosis based on the ICD-9-CM code was an acceptable and reliable method, although it may not be specific enough. Therefore, we have modified the limitation and acknowledged this shortcoming.

Modified:

“6. Limitations of the study

This is a single-center retrospective study. PLR can be time-varying, which was not fully captured in the current analysis. For logistic regression with the univariate and multivariate analyses, models without adjustment for co-variates may be biased due to uncertainty or lack of confounders. For example, the interaction or non-linearity for the relationship between covariates and outcome were not be considered in this manuscript. To solve this problem, ensemble modeling can address non-linearity automatically without pre-specification in future research. Moreover, the disease definition of the MIMIC III database is based on the ICD-9-CM code. Therefore, some important information is lacking in the database, including cardiac color Doppler ultrasound results.”

[1] Johnson AE, et al. MIMIC-III, a freely accessible critical care database. Sci Data. 2016;3:160035. doi: 10.1038/sdata.2016.35.

5.Line 93-95 – This statement can be interpreted to mean that since there was little missing data, then it was omitted. But I believe the authors are trying to say there was little missing data so it was included in analysis or something like this? Needs to be clarified

Response: Thanks for your comments. For better understanding, we changed the sentence to “Since there was little missing data (<1.5%), it was omitted in the further investigation”. We hope the changes meet your requirements. Thank you.

  1. Table 1 – the demographics included need to include female gender if includes male. In addition, age is not as helpful as gestational age at birth. Heart rate is also not as clinically helpful given it is not clear where this heart rate comes from (just at birth? On average over duration of hospital stay?) and can fluctuate depending on the patients status. If a hemodynamic marker is to be included, mean arterial pressure (MAP) should be included. The Comorbidities contradict the exclusion criteria as hyaline membrane disease is a pulmonary dysfunction disease. Need clarification on what a “pulmonary circulation disorder” is.

Response: Thanks for your comments. The value of female gender has been added in the table. The definition of disease in the database was mainly based on the ICD-9-CM code, which strictly determined by trained doctors. Some variables such as echocardiography, gestational age, MAP, pre-hospital medication or detailed medical history may be unavailable or contain many missing values. The enrolled patients did not have hyaline membrane disease at diagnosis, but may have acquired pulmonary diseases, such as apnea caused by inflammation. In the patient's baseline form, it can be found that most of the patients' inflammatory markers are abnormal. Therefore, even without hyaline membrane disease, the possibility of other pulmonary complications exists. However, we have selected all the indicators currently available in the database, which may not have much practical clinical use, and our statistical results have also confirmed that there was no statistical significance.

  1. Table 3 – this is probably the crux of the paper in graphic form, however it is confusing. The percentages are not clarified in regards to what they mean (% of population? Mortality?) and the trends are not clear. Either need to take away percentages or clarify in the table description.

Response: Thanks for your comments. The percentages indicated the ratios of survivors or nonsurvivors in each Quartiles in total patients of total patients. For better understanding, we modified the table description as followed:

“LMR, lymphocyte-to-monocyte ratio; NLR, neutrophil-to-lymphocyte ratio; PLR, platelet-to-lymphocyte ratio; Q, Quartiles of LMR, NLR and PLR. %, percentage of survivors or nonsurvivors in each Quartiles in total patients of total patients. For each indicator, the data of upper column was for the survivors, and the below for nonsurvivors.” We hope the changes meet your requirements.

  1. Line 201 – grammatical error, need to take out second “was”

Response: Thanks for your comments. The sentences have been revised as the suggestion and highlighted in manuscript.

  1. Line 206 – “hospital mortality was moderately good”. How was moderately good predictive ability determined?

Response: Thanks for your comments. According to the recent paper (PMID: 34633993 DOI: 10.1213/ANE.0000000000005773) [1], AUC was considered excellent discrimination when ≥ 0.90, good discrimination when between 0.8-0.89, moderate discrimination when between 0.7-0.79, poor discrimination when between 0.6-0.69, very poor discrimination when between 0.51-0.59. Therefore, we modified our results as “The prognostic predictive ability of models combining PLR with other variables for hospital mortality was good”. The modified was highlighted in the manuscript.

[1] Patrick Schober, Edward J Mascha, Thomas R Vetter. Statistics From A (Agreement) to Z (z Score): A Guide to Interpreting Common Measures of Association, Agreement, Diagnostic Accuracy, Effect Size, Heterogeneity, and Reliability in Medical Research. Anesth Analg. 2021;133(6):1633-1641. doi: 10.1213/ANE.0000000000005773.

Round 2

Reviewer 2 Report

Good for publication